# Radiologic Definition of Sarcopenia in Chronic Liver Disease

**DOI:** 10.3390/life11020086

**Published:** 2021-01-25

**Authors:** Chul-min Lee, Bo Kyeong Kang, Mimi Kim

**Affiliations:** Department of Radiology, College of Medicine, Hanyang University, Seoul 04763, Korea; 2200193@hyumc.com (C.-m.L.); msbbogri@naver.com (B.K.K.)

**Keywords:** sarcopenia, chronic liver disease, radiologic evaluation, DXA, CT, ultrasonography, MRI

## Abstract

Sarcopenia is prevalent in patients with chronic liver disease, and affected patients tend to have worse clinical outcomes and higher mortality. However, relevant analyses are limited by heterogeneity in the definition of sarcopenia and in the methodological approaches in assessing it. We reviewed several radiologic methods for sarcopenia in patients with chronic liver disease. Dual energy X-ray absorptiometry (DXA) can measure muscle mass, but it is difficult to evaluate muscle quality using this technique. Computed tomography, known as the gold standard for diagnosing sarcopenia, enables the objective measurement of muscle quantity and quality. The third lumbar skeletal muscle index (L3 SMI) more accurately predicted the mortality of subjects than the psoas muscle index (PMI). Few studies have evaluated the sarcopenia of chronic liver disease using ultrasonography and magnetic resonance imaging, and more studies are needed. Unification of the measurement method and cut-off value would facilitate a more systematic and universal prognosis evaluation in patients with chronic liver disease.

## 1. Introduction

Sarcopenia is defined as reduced skeletal muscle mass and reduced muscle functionality [1]. Primary sarcopenia is a naturally occurring phenomenon with aging; however, when its severity is due to chronic illness beyond what can be justified by aging alone, it is called secondary sarcopenia [2]. It is also prevalent in patients with chronic liver disease. Sarcopenia is associated with a higher incidence of hepatocellular carcinoma, significant liver fibrosis, hepatic encephalopathy after transjugular intrahepatic portosystemic shunt (TIPS), high list mortality, postoperative mortality, and complications in patients with end-stage liver disease [3,4,5]. Sarcopenia is recognized as a disease entity in the International Classification of Disease (ICD-10) [6].

Despite growing research on sarcopenia, progress is hampered by the lack of unified definitions. Several imaging modalities, such as dual energy X-ray absorptiometry (DXA), ultrasonography (US), computed tomography (CT), and magnetic resonance imaging (MRI) have made it possible to use body composition assessments for patients with chronic liver disease. Measuring muscle mass with several forms of imaging is less affected by acute illness or cognitive dysfunction compared with measuring strength or physical performance and can be objectively employed in clinical practice [7]. This review evaluates studies related to the definitions of, and methodological approaches to diagnose, sarcopenia in adults with chronic liver disease.

## 2. Methods to Evaluate the Quantity and/or Quality of Skeletal Muscle Mass in Chronic Liver Disease

### 2.1. DXA

Measurements of skeletal mass are provided by DXA, which allows for the quantification of three body compartments (bone mass, fat mass, and bone fat-free mass (or lean mass)) based upon the differential tissue attenuation of X-ray photons (Figure 1).

Lean mass includes muscle and other components such as skin, tendons, and connective tissues. DXA is biased by the fluid overload that is frequently present in decompensated cirrhosis [8]. Lean mass, moreover, cannot differentiate muscle from water and can be overestimated in patients with cirrhosis who are experiencing water retention, such as ascites and peripheral edema. To minimize confounding by the fluid, both appendicular (arms and legs) and upper limb lean masses have been proposed as tools to analyze the muscle mass in cirrhosis [9]. The abdominal compartment is excluded from appendicular skeletal muscle measurements. This approach may not be influenced by ascites and thus may be useful for skeletal muscle mass estimations in patients with cirrhosis [10]. Low lean mass and appendicular limb lean mass assessed by DXA predict the risk of muscle weakness and limited mobility but are not associated with significant clinical outcomes, such as mortality [11]. Muscle mass is correlated with body size. The International Society for Clinical Densitometry recommended adjusting these values to body size. Muscle mass can be adjusted by height [9], weight [12], or body mass index (BMI) [13,14], although it is not known which of these methods is superior. The appendicular skeletal mass index (ASMI), adjusted by height, is calculated as follows:$$\mathrm{Appendicular}\;\mathrm{skeletal}\;\mathrm{mass}\;\mathrm{index}\;\left(\mathrm{ASMI}\right)=\frac{Appendicular\;skeletal\;mass\;\left(Kg\right)}{height\left(m\right)\times height\left(m\right)}$$

Lindqvist et al. noted that ASMI showed similar results to SMI (skeletal muscle index) based on CT (γ = 0.66, *p* < 0.001), but the correlation between the lean mass index and SMI was weaker (γ = 0.29, *p* = 0.035) and falsely high in patients with ascites in the image analysis of patients before liver transplantation (LT) [15]. To date, several consensus definitions using DXA have been proposed by the International Working Group on Sarcopenia (IWGS), Asian Working Group for Sarcopenia (AWGS), and European Working Group on Sarcopenia in Older People (EWGSOP) (Table 1) [1,16,17]. The cut-off was based on the general population and should also be validated in patients with chronic liver disease (Table 1).

The limitation of DXA is its inability to assess muscle quality in contrast to CT and MRI. DXA cannot quantify intramuscular adipose tissue within and around muscles, and DXA-measured ASMI has shown only moderate correlation with SMI based on CT (γ = 0.41–0.66), which is considered to be the gold standard for estimating muscle mass in research [15,18]. Despite these limitations, DXA is commonly used in primary care in both clinical and research settings, as it has the advantages of being safe, inexpensive, and reproducible, as well as providing low radiation exposure.

### 2.2. US

US has also been applied to measure muscle size and myosteatosis. US is easy, inexpensive, and uses portable equipment, allowing it to be performed at the bedside, with no harm to the patient from radiation. US-based measurements have shown positive correlations with DXA-, CT-, and MRI-based measurements [19,20,21]. However, the reference standards used in each study were different, and there remains no standardization of the measurement technique. Perkisas et al. provided a standardization method for assessing appendicular muscle with US [22], but there was no clear cut-off for appendicular muscles in diagnosing sarcopenia. Only a few studies have evaluated the sarcopenia of chronic liver disease using US (Table 2).

Tandon et al. demonstrated that a nomogram based on body mass index and right thigh muscle thickness can identify male and female patients with sarcopenia defined by cross-sectional CT [23]. Hari et al. showed the possibility of using US for the evaluation of sarcopenia in patients with chronic liver disease by measuring the diameter of the right psoas muscle [24] (Figure 2). The authors reported that the success rate of measuring the psoas muscle was 72%. Technical failure was due to a poor sonic window with a high abdominal circumference or the presence of ascites. US can be used as a screening tool for sarcopenia in situations where other imaging methods are not available because of their radiation, high cost, and lack of portability.

### 2.3. CT

CT imaging is increasingly used as the gold standard to quantify skeletal muscle mass along with MRI and constitutes a good resource for the objective identification of sarcopenia. However, CT suffers from the disadvantage of radiation exposure. Nevertheless, patients with chronic liver disease frequently receive CTs to evaluate hepatocellular carcinoma or the complications of portal hypertension.

Generally, the cross-sectional areas of the psoas or abdominal muscle mass of the third (L3) or fourth (L4) lumbar vertebra levels that are not affected by activity are used. Although the cross-sectional area of the level of the umbilicus has previously been used [26], this area may be measured at different levels of the vertebra. Moreover, the umbilicus may become flat and unable to be found on axial CTs in patients with ascites or obesity [43]. Measuring should be conducted at the cross-sectional area of the skeletal muscles at the level of L3, which accurately represents the whole-body skeletal muscle mass [44]. At the L3 transverse section, muscle groups include the rectus abdominis, transverse abdominis, internal and external obliques, quadratus lumborum, psoas major and minor, and erector spinae. These measurements were computed by summing the tissue pixels and multiplying them by the pixel surface area. A threshold range of 29 to 150 HU is commonly used to define the muscle (Figure 3).

Most of the measured area is corrected with height and sometimes with BMI [45]. Height correction is necessary to determine the relative muscle mass because of the linear relationship between skeletal muscle and height [46]. The SMI is calculated as follows:$$\mathrm{Skeletal}\;\mathrm{muscle}\;\mathrm{index}\;(\mathrm{SMI})\;\mathrm{of}\;L3=\frac{Total\;muscle\;area\;of\;L3\;level\;\left({\mathrm{cm}}^{2}\right)}{height(m)\;\times height\left(m\right)}$$

Occasionally, the psoas muscle is selected from other skeletal muscles of the region because it is located centrally, is easily identified, and is not directly affected by abdominal distension in the presence of ascites. There are several methods for measuring the area of the psoas muscle:

(1) Transverse psoas muscle thickness (TPMT), where the greatest transverse diameter of the psoas muscle runs perpendicular to the long axis (anterior–posterior oblique) of the psoas muscle diameter [27]. The results were normalized to body height and are shown as mm/m:$$\mathrm{TPMT}/\mathrm{height}\;=\;\frac{Transverse\;psoas\;muscle\;thickness\;\left(mm\right)}{height\;\left(m\right)}$$

(2) The psoas muscle index (PMI), where the estimated or measured psoas muscle area is adjusted by height [33]. The estimated psoas muscle area is the sum of the product of the long axis and the short axis of the iliopsoas muscles on both sides [2]. The estimated psoas muscle area and PMI are calculated as follows:Estimated psoas muscle area = (a × b) + (c × d)
where a is the TPMT of the right psoas muscle, b is the longitudinal psoas muscle thickness (LPMT) of the right psoas muscle, c is the TPMT of the left psoas, and d is the LPMT of the left psoas muscle.
$$\mathrm{Psoas}\;\mathrm{muscle}\;\mathrm{index}\;(\mathrm{PMI})\;\mathrm{of}\;L3=\frac{Total\;psoas\;muscle\;area\;of\;L3\;level\;\left({\mathrm{cm}}^{2}\right)}{height(m)\;\times \;height{\left(m\right)}_{}}$$

A consensus definition using PMI was proposed by the Japan Society of Hepatology guidelines for sarcopenia. This definition includes a cut-off value for the measured psoas muscle area of 6.36 cm^2^/m^2^ in males and 3.92 cm^2^/m^2^ in females and a cut-off value for the estimated psoas muscle of 6.0 cm^2^/m^2^ in males and 3.4 cm^2^/m^2^ in females [2] . Other studies used the cut-off value derived from mortality or morbidity [29,30,39] or from the sex-specific lowest quartile or fifth percentile of the subjects [32,33].

However, there is no evidence confirming that the cross-sectional area of the psoas muscles has a good correlation with the whole-lumbar or the whole-body muscle volume. Among 396 patients with end-stage liver disease, PMI was less likely to predict mortality than SMI in male patients. Male patients who died had a lower SMI but not a lower PMI compared to male patients who were alive [39]. Therefore, the entire skeletal muscle at the L3 vertebra level should be measured.

The attenuation of muscle was additionally measured to evaluate muscle quality. Low muscle attenuation, referred to as myosteatosis, indicates increased intramuscular fat content, which contributes to muscle weakness independent of the age-associated loss in muscle mass [47,48]. Wang et al. showed that myosteatosis, but not muscle mass, is related to mortality in a study of 292 patients with end-stage liver disease [36].

The same cut-off value cannot be uniformly applied because the muscle mass varies according to age, sex, BMI, and ethnicity. There is wide heterogeneity in the cut-off values of SMI used to defined sarcopenia in chronic liver disease: A sex-specific cut-off value derived in patients with solid tumors related to mortality [34,35,49,50], a cut-off value according to BMI [36,37], a cut-off according to age [33], a cut-off defined from control subjects [33], and a cut-off from subjects using the sex-specific lowest quartile/tertile [32,38,40]. The North American Working group on Sarcopenia in Liver Transplantation suggested that the definition of sarcopenia in patients with end-stage liver disease waiting for LT should be defined as SMI less than 50 cm^2^/m^2^ for males and less than 39 cm^2^/m^2^ for females at the L3 level [7]. In their study, there were no statistically significant differences in SMI according to ethnicity.

### 2.4. MRI

MRI can also be used to evaluate muscle quantity and quality. MRI provides high resolution and permits the separation and quantification of muscle compartments and fat distribution. It can be used to evaluate detailed anatomical changes of muscle, including muscular atrophy, fatty degeneration, and edema. In contrast to CT, MRI carries no risk of ionizing radiation exposure or kidney injury due to iodine contrast media administration. MRI can be performed without the administration of contrast media because of its high soft-tissue contrast and multiparametric characteristics.

With advances in MRI techniques, the assessment of the chemical composition of tissue has also become possible. MR spectroscopy (MRS) is a representative MR technique that assesses the chemical composition of tissue [51,52]. DIXON-based MRI is a recent MRI technique challenging MRS by using a chemical shift to enable the selective reconstruction of fat signal- and water signal-only images [53,54]. Quantitative analysis using DIXON-based MRI showed an excellent correlation in MRS, which outperformed visual assessment in the detection of muscle fat content [55]. These fat quantification MRI techniques could be applied to the evaluation of sarcopenia in terms of muscle quality assessments in patients with chronic liver disease (Figure 4).

Compared with CT, which is the gold standard for the evaluation of sarcopenia in several guidelines, MRI has shown similar performance in the evaluation of sarcopenia in healthy subjects [56,57]. Traditionally, MRIs have been used for muscle quality and quantity evaluations of neuromuscular disorders [58,59,60]. Furthermore, an MRI-only assessed sarcopenia is an important prognostic factor in many types of cancer, including breast cancer, head and neck cancer, and colorectal cancer [61,62,63]. However, this method remains in the early stages for studying sarcopenia in chronic liver disease. There are two studies on the adverse effects of decreased muscle mass diagnosed by TPMT-adjusted height and the spinae muscle area (Table 2) [41,42]. To our knowledge, there has been no study on the adverse effects of reduced muscle mass at the L3 level, and there is also a lack of research articles on myosteatosis in patients with chronic liver disease, so further studies and validations are needed. The concern regarding the high cost of MRI can be considered negligible because patients with chronic liver disease are frequently subjected to MRIs for several clinical reasons.

## 3. Considerations for the Radiologic Evaluation of Sarcopenia

The radiologic evaluation of sarcopenia has the advantage of being able to measure muscle mass objectively and quantitatively. CT and MRI, which are considered to be gold standards for muscle mass measurements, showed high interobserver agreement with Pearson’s correlation coefficient [39,63]. The measurements of skeletal muscle mass on CT and MRI were also found to be interchangeable. Park et al. showed very good agreement between CT and MRI measurements of skeletal muscle mass at the level of the L3 vertebra (the ICC of reader 1 was 0.928 and that of reader 2 was 0.853) [64]. This result is also consistent with studies carried out at the level of the superior mesenteric artery (mostly the first lumbar vertebra) and the level of the third cervical vertebra (C3) [57,63]. In addition, it is possible to measure skeletal muscle retrospectively because most patients with chronic liver disease frequently undergo radiologic evaluations.

In previous studies, different cut-off values were applied according to sex, etiology of the disease, ethnicity, and the modality used. The SMI values of male patients were significantly higher than those of female patients; however, the frequency of sarcopenia among male patients was higher than that in female patients when sex-specific cut-offs were applied [31,35,39]. Previous studies have examined the association between non-alcoholic fatty liver disease (NAFLD) and hepatitis B or C viral cirrhosis with sarcopenia using DXA as the assessment tool [12,13,25,65]. A study from Korea reported the association of NAFLD with sarcopenia using CT [45], with a cut-off value defined as 1 standard deviation below the sex-specific mean value for a young healthy population: 8.37 cm^2^/(kg/m^2^) for men and 7.47 cm^2^/(kg/m^2^) for women. After the publication of the EWGSOP guidelines, some studies were designed to determine specific cut-off values for sarcopenia assessment using CT for Japanese and Asian adults. Two studies provided PMI cut-off values of 3.74 and 6.36 cm^2^/m^2^ for men and 2.29 and 3.92 cm^2^/m^2^ for women, respectively, based on Japanese liver donor data. The authors suggested that that the cut-off values in Western studies could be different from the actual values in Asian populations due to differences in body sizes, lifestyles, and ethnicities [25,66]. Further studies to define sarcopenia should be conducted according to ethnicity and the etiology of hepatic disease.

There is a need for standardized CT, as CT parameters such as tube potential, the use of a contrast agent, and slide thickness also affect the assessment of skeletal muscle. A reduction in tube potential from 140 to 80 kV leads to a 5.2% decrease in SMI [67], and the use of contrast media overestimates the average SMI by up to 2.8% [68]. Differences in slice thickness of 10 and 2 mm can result in a 1.9% smaller SMI [68]. Contrast enhancement, moreover, strongly influences the value of skeletal muscle density [69,70].

Moreover, radiological assessment does not always reflect strength or physical performance [71,72,73,74]. However, further research is needed to determine which parameters of muscle strength and physical performance are complemented by radiological assessment. In 2018, EWGSOP recommended that if a patient has low muscle strength, is defined as probable sarcopenia, and has low muscle quantity or quality, they can be diagnosed as sarcopenia [1] because it is recognized that strength is better than muscle mass in predicting adverse outcomes. Sinclair et al. showed that the model for the end-stage liver disease (MELD)-handgrip strength bivariate Cox model is superior to the MELD-CT muscle Cox model (*p* < 0.001) in predicting mortality [18]. For muscle strength, the use of a handheld dynamometer is a valid and reliable method with high interrater and intrarater reliability. Further, the short physical performance battery and gait speed provide good measurement properties for the assessment of physical performance [1,75].

## 4. Conclusions

The evaluation of sarcopenia is crucial in patients with chronic liver disease, as well as other chronic illness, because sarcopenia is one of the important prognostic factors [4,5]. Overall, CT and MRI are considered the gold standard for evaluating sarcopenia and are frequently performed in patients with chronic liver disease for the evaluation of hepatocellular carcinoma or complications of portal hypertension. CT shows excellent performance in estimating the quality and quantity of muscle, and many studies have reported variable measurement methods and cut-off values in patients with chronic liver disease. MRI could be a competent imaging modality for muscle quality evaluation by measuring intramuscular fat content with MRS or DIXON-based MRI, as well as muscle mass by measuring the area, which requires further validations in chronic liver disease. DXA is a reliable alternative for clinical use when a CT scan is not clinically indicated or available. Unification of the measurement method and cut-off value would facilitate more systematic and universal prognosis evaluations in patients with chronic disease.

## Figures and Tables

**Figure 1 life-11-00086-f001:**
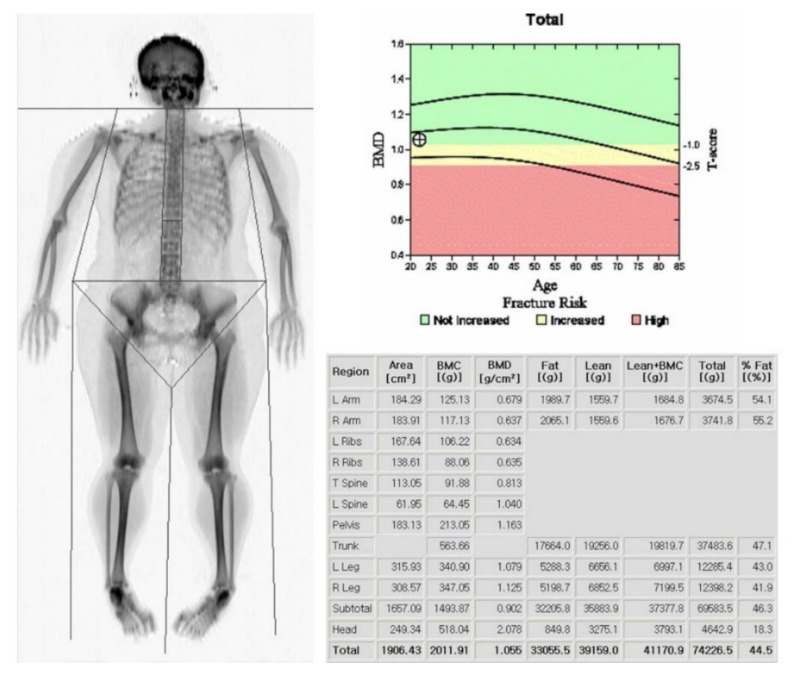
Dual energy X-ray absorptiometry (DXA) scan with body composition analysis. The subject’s height was 1.59 m, and the lean and appendicular lean masses were 39.1 and 16.6 kg, respectively. The Appendicular Lean Mass Index (ALMI) was calculated as 6.6 kg/m^2^.

**Figure 2 life-11-00086-f002:**
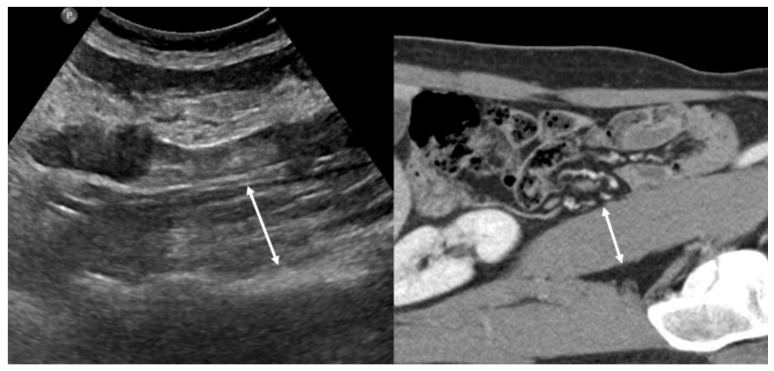
Left side: Example of an ultrasound measurement of the psoas muscle diameter. Right side: the corresponding CT image.

**Figure 3 life-11-00086-f003:**
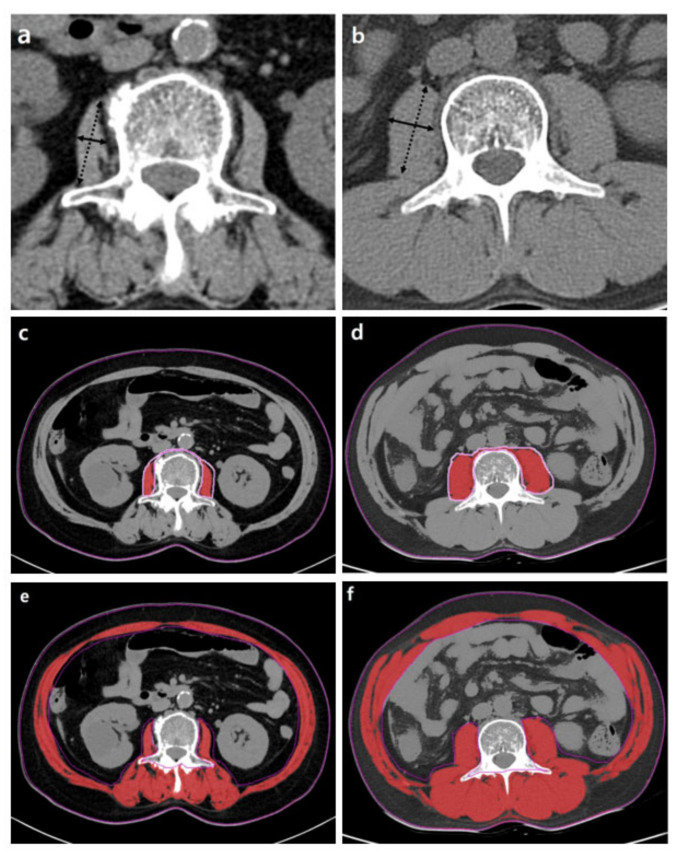
Abdominal CT images taken at the third lumbar vertebra with cirrhosis applied to quantify transverse psoas muscle thickness (TPMT), psoas muscle, and total muscle areas. The psoas muscle in panels (**c**,**d**) and total muscle area in panels (**e**,**f**) are colored in red. Panels (**a**,**c**,**e**) present a female patient with low TPMT (5.9 mm/m), psoas muscle index (PMI) (2.3 cm^2^/m^2^), and skeletal muscle index (SMI) (35.7 cm^2^/m^2^). Panels (**b**,**d**,**f**) present a male patient with high TPMT (23.3 mm/m), PMI (6.8 cm^2^/m^2^), and SMI (51.6 cm^2^/m^2^). The mean density of muscle in (**c**–**f**) is 34.7, 38.2, 26.2, and 36.6 HU, respectively.

**Figure 4 life-11-00086-f004:**
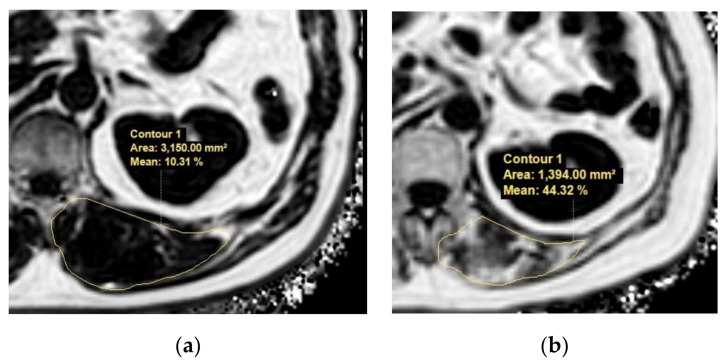
Example MR images (fat fraction) of cross-section of the spinal muscles at the Superior Mesenteric Artery level in two patients with low fatty degeneration (**a**) and high fatty degeneration (**b**). (**a**) The area and fat fraction of the left spinal muscle were 3150 mm^2^ and 10.3%. (**b**) The area and fat fraction of the left spinal muscle were 1394 mm^2^ and 44.3%. An axial three-dimensional multi-echo-modified Dixon gradient echo sequence was used for the MRI. Imaging parameters for the sequence were as follows: six Echo Time (TE)s (first TE shortest automatic (0.9–1.2ms), delta TE 0.8–1.01ms); Repetition Time (TR) shortest automatic (5.8–6.3 ms); flip angle =3; field of view = 35 × 35 cm^2^; 3 mm slice thickness with no gap; matrix size = 300 × 300; number of slices = 60; scan time = 14.1 s; parallel imaging SENSE factor = 2; number of signal average = 1. The sequence automatically produces water, fat, fat fraction, R2*, and T2* maps.

**Table 1 life-11-00086-t001:** Recommended cut-off values for muscle mass using dual energy X-ray absorptiometry (DXA).

	IWGS, 2011	AWGS, 2014	EWGSOP, 2018
ASMI	7.23 kg/m^2^ for men 5.67 kg/m^2^ for women	7.0 kg/m^2^ for men 5.4 kg/m^2^ for women	7.0 kg/m^2^ for men 5.5 kg/m^2^ for women

**Table 2 life-11-00086-t002:** Summary of studies investigating radiologic sarcopenia in chronic liver disease.

Author	Subjects (M/F)	Mean Age (Years)	Country /Ethnicity (%)	Etiology (%)	Body Composition Methods	Definition of Sarcopenia	Prevalence	Adverse Effects
Lee 2016 [13]	2,761 (1,240:1,520)	55.8	Korea	NAFLD	DXA, ASMI (ASM adjusted by BMI)	0.789 for men, 0.521 for women	12.2% M: 10.6% F: 13.5%	Sarcopenia was associated with significant liver fibrosis (OR = 0.52–0.67, *p* < 0.01)
Han 2018 [25]	506 (258:248)	50.7	Korea	Chronic hepatitis B	DXA, ASMI (ASM adjusted by BMI)	0.89 for men, 0.58 for women	24.9%	Sarcopenia was associated with significant liver fibrosis (OR = 2.01–3.62, *p* < 0.05)
Belarmino 2018 [10]	144 (144:0)	54.0	Brazil	Cirrhosis 59 alcoholic 20 viral 12 cryptogenic 9 other	DXA, ASMI (ASM adjusted by height)	Tertile of patients (<7 Kg/m^2^) plus nondominant handgrip strength < 25 Kg	13.2%	Patient with sarcopenia showed higher mortality (*p* < 0.001)
Sinclair 2019 [9]	420 (420:0)	55.4 (median)	Australia	ESLD 28.3 HCC 24.3 HCV, 12.6 alcoholic 10.2 PSC, 6.2 NAFLD 18.3 other	DXA, ALMI (ALM adjusted by height)	<7.26 Kg/m^2^	30.9%	ALM and lean mass of the arms were inversely associated with mortality (HR = 0.78, *p* = 0.03 for ALM and HR = 0.37, *p* = 0.02 for lean mass of the arms)
Hari 2019 [24]	75 (39:36)	63	Slovenia	Cirrhosis 67 alcohol 15 NAFLD 18 other	US, (1) psoas to height ratio (right psoas muscle diameter divided by height) (2) right PMI (π · psoas^2^/height^2^)	NA	NA	Psoas to height ratio was related to hospitalization (HR = 0.72, *p* < 0.001) and mortality (HR = 0.82, *p* = 0.022) PMI was related to hospitalization (HR = 0.88, *p* < 0.001) and mortality (HR = 0.93, *p* = 0.017)
Durand 2014 [26]	562 (455:107)	53	88 Caucasian 10 Afrian 2 Asian	ESLD 42 alcohol 15 HBV 30 HCV 5 biliary disease 8 others	CT at level of umbilicus, TPMT/height	NA	NA	TPMT/height was an independent predictive factor of waiting list mortality (HR = 0.87, *p* = 0.001).
Kim 2014 [27]	65 (41:24)	55	Korea	Cirrhosis 56.9 alcohol 26.1 viral 9.3 mixed 7.7 other	CT, L4, TPMT/height	NA	NA	Mortality with TPMT/height ≥14 mm/m was higher than TPMT/height <14 mm/m (HR = 5.4, *p* < 0.001)
Krell 2013 [28]	207 (129:78)	51.7	81.2 White 11.1 African American 7.7 Other	Patients with LT 26.1 HCV 25.1 HCC 14.5 alcohol 34.3 other	CT, L4, total psoas area	Sex-specific tertiles	33%	Patients in the lowest tertile had a greater than 4-fold higher change in developing a severe infection in comparison with patients with in the highest tertile (OR = 4.6, 95 % CI = 2.25–9.53)
Tateyama 2020 [29]	99 (61:38)	70	Japan	Cirrhosis 52.5 HCV 18.2 HBV 29.3 other	CT, L3, PMI, SMI	<4.3 cm^2^/m^2^	35.4%	The incidence of minor hepatic encephalopathy was frequent in patients with lower PMI than higher PMI (*p* = 0.001)
Hou 2020 [30]	251 (129:122)	62.6	China	Cirrhosis 38.3 viral 15.9 alcohol 8.4 autoimmune 37.5 other	CT, L3, PMI	Male: <3.5 cm^2^/m^2^ Female: <2.6 cm^2^/m^2^	NA	PMI was associated with 3-year mortality in male (HR = 0.673, *p* = 0.020) and in female (HR = 0.586, *p* = 0.013).
Masuda 2013 [31]	204 (103:101)	54	Asian	Patients with LT 12.7 HBV, 50.5 HCV 13.2 PBC 4.9 alcoholic 18.6 other	CT, L3, psoas muscle area	Lowest 5th percentile <800 cm^2^ for men <380 cm^2^ for women	47.1% M: 58.3% F: 35.6%	Sarcopenia was an independent predictor of postoperative sepsis (HR = 5.31, *p* = 0.009)
Kalafateli 2017 [32]	232 (162:70)	53	76.3 Caucasian 5.6 African 16.8 Asian 1.3 Other	Patients with LT 20.3 autoimmune 34.9 viral 23.7 alcohol 21.1 other	CT, L3, PMI	Lowest quartile <340 mm^2^/m^2^ for male <264 mm^2^/m^2^ for female	24.6% M: 24.7% F: 24.3%	L3-PMI was associated with longer hospital stay (OR = 0.996, 95% CI = 0.9940.999) and 1-year mortality (OR = 0.996, *p* = 0.05).
Tsien 2014 [33]	53 (41:12)	56.9	USA	Patients with LT 41.5 viral 22.6 mixed 7.5 NASH 28.3 other	CT, mid L4 level, PMI and muscle attenuation.	5^th^ percentile of control subjects Male, <50 years old: <12.27 cm^2^/ m^2^ Male, >50 years old: <10.12 cm^2^/m^2^ Female: <50 years old: <10.47 cm^2^/m^2^ Female: >50 years old: <10.33 cm^2^/m^2^	62.3% at pre-LT 86.8% at post-LT	Pre-LT sarcopenia increased mortality (*p* = 0.06). Continued sarcopenia in post-LT showed a trend of higher mortality (*p* = 0.08)
Tanai 2016 [34]	149 (82:37)	65	Japan	Cirrhosis 53.0 HCV 22.8 alcohol 6.7 HBV 17.4 other	CT, L3 level, SMI	<52.4 cm^2^/m^2^ for men <38.5 cm^2^/m^2^ for women	63% M: 76% F: 48%	Relative change in skeletal muscle area (<3.1%) was associated with mortality (HR = 2.73, *p* < 0.001).
Nardelli 2017 [35]	46 (34:12)	58.6	Italy	Patients with cirrhosis received TIPS	CT, L3-4 disc space, SMI	Same as above	57%	Sarcopenia was associated with development of hepatic encephalopathy after TIPS (HR = 3.13, *p* < 0.001)
Wang 2016 [36]	292 (193:99)	61	55 Non-hispanic white 5 Black 26 Hispanic 8 Asian 6 Other	ESLD 60 HCV 11 alcohol 8 NAFLD 10 cholestatic 12 other	CT, L3, SMI and muscle attenuation	Muscle mass <43 cm^2^/ m^2^ for men with BMI <25 Kg/m^2^ <53 cm^2^/m^2^ for men with BMI ≥25 Kg/m^2^ <41 cm^2^/m^2^ for women with any BMI Reduced muscle attenuation <41 HU for BMI <25 Kg/m^2^ <33 HU for BMI ≥25 Kg/m^2^	Sarcopenia: 38% Poor muscle quality: 50%	Muscle quality was associated with waitlist mortality (HR = 0.77, *p* = 0.02), but muscle mass was not (HR = 0.91, *p* = 0.35).
Montano-Loza 2016 [37]	678 (457:221)	57	Canada	Cirrhosis 40 HCV 23 alcohol 14 NASH and cryptogenic 8 autoimmune 6 HBC 1 other	CT, L3, SMI and muscle attenuation	Same as above	Sarcopenia: 43% Poor muscle quality: 52.1%	Sarcopenia (HR= 2, *p* < 0.001) and poor muscle quality (HR = 1.42, *p* = 0.04) were associated with mortality.
Carey 2017 [38]	396 (277:119)	58	71 Non-hispanic white 5 Black 11 Hispanic 8 Asian 7 Other	Patients with LT 48 HCV 17 alcohol 12 NAFLD 5 HBV 17 other	CT, superior aspect of L3, SMI	<50 cm^2^/ m^2^ for men <39 cm^2^/ m^2^ for women	44.9%	Sarcopenia was associated with waitlist mortality (HR = 0.95, *p* < 0.001)
Ebadi 2018 [39]	353 (246:107)	56	USA	ESLD	CT, L3, PMI and SMI	SMI <50 cm^2^/m^2^ for men <39 cm^2^/m^2^ for women PMI (cut-off calculated from subjects) <5.1 cm^2^/m^2^ for men <4.3 cm^2^/m^2^ for women	Sarcopenia by SMI: 47% M: 51% F: 36%	In women, both low SMI and PMI were predictors of mortality (HR = 2.05, *p* = 0.05 for low SMI and HR = 2.47, *p* = 0.01 for low PMI). In men, low SMI was only significant predictor of mortality (HR = 2.46, *p* = 0.002).
Fujiwara 2015 [40]	1,257 (828:429)	68.6	Japan	Patients with HCC 71.2 HCV 11.3 HBV 17.5 other	CT, L3, SMI and muscle attenuation	Muscle mass <36.2 cm^2^/m^2^ for men <29.6 cm^2^/m^2^ for women Reduced muscle attenuation <44.4 HU for men <39.3 HU for women	Sarcopenia: 11.1% M: 11.6% F: 10.0% Poor muscle quality: 85.0%	Sarcopenia (HR = 1.52, *p* = 0.001) and poor muscle quality (HR = 1.34, *p* = 0.020) were significant predictors of survival.
Beer 2020 [41]	265 (164:101)	54	Austria	CLD 21 HCV 19 alcohol 9 HBV	MR, L3, TPAM/height	<12mm/m for men <8mm/m for women	27.2%	Sarcopenia was risk factor for mortality (HR = 2.76, *p* = 0.045) for compensatory advanced chronic liver disease.
Praktiknjo 2018 [42]	116 (69:47)	59	EU	Patients with cirrhosis received TIPS 62.9 alcohol 15.5 viral 21.6 other	MR, at level of SMA, spinae muscle area and fat-free muscle area	Sex specific cut-off from subjects Total muscle <3523 mm^2^ and fat-free muscle <3197 mm^2^ for men Total muscle <3153 mm^2^ and fat-free muscle <2895 mm^2^ for women	44.0%	Persistence of sarcopenia after TIPS is associated with mortality (HR = 5.62, *p* = 0.001)

## Data Availability

No new data were created or analyzed in this study. Data sharing is not applicable to this article.

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
