# Peer review of "Radiologic Definition of Sarcopenia in Chronic Liver Disease"

_life, 2021, doi:10.3390/life11020086_

Round 1
Reviewer 1 Report
This is an overview of the current methods to estimate muscle mass in patients with chronic liver disease. The reference list is rather good but the presentation needs more polishing, especially the section on MRI, which needs a lot of work. I would also like to see more discussion on differences among populations and gender. Asian, European, American, African and Afro-american will most probably be quite different. When revising the manuscript I suggest first giving the manuscript a major overhaul to make the presentation more didactic. Include figures of DXA, a thresholded CT of L3 level and MRI images. Include the equations and CT images showing how to make the measures ASMI, SMI, TPMT and PMI. Describe how adjustment for body height is done. Finalize the revision by linguistic polishing by a native, professional copy editor immediately prior to re-submission as there are several linguistic errors that are not so easy to correct unless being native speaking. Below I have given some specific comments. Do note that the list is not complete – those are examples.
” Low muscle mass is not always equivalent to low muscle strength, however, measuring muscle mass with several imaging is least affected by acute illness or cognitive dysfunction and can be objectively measured in clinical practice [5].” The sentence does not make sense.
The abbreviation DEXA is not wrong, but DXA is more commonly used.
“Measures of skeletal mass are provided by DEXA, which allows for the quantification of three body composition (bone mass, fat mass, and bone fat free mass (or lean mass)) based upon differential tissue attenuation of x-ray photons.” Should be ”compartments”, not ”compositions”.
“The limb muscle is affected by activity, psoas, and abdominal muscle mass of the third lumbar (L3) or fourth lumbar (L4) vertebra level that are not affected by activity are used.” Two messages in one sentence. Reformulate.
“Although the level of the umbilicus has been used as a mark for the location of the muscle area [19], these measures may be recorded at different levels in these patients with ascites or obesity.” Unclear, explain it better.
“But, there is no evidence confirming that the cross-sectional area of the psoas muscles has a good correlation with the whole lumbar or the whole body muscle areas.” I am not aware of any technique to measure whole body muscle area. I guess that you mean volume?
“Deceased male 100 patients had lower SMI, but not PMI compared with non-deceased male patients [23].” Some spelling mistakes are funny. “Deceased” means “dead”. You probably mean “diseased”.
As shown by Morsbach et al, the parameters used when performing the CT scanning do also affect the assessment of skeletal muscle. A reduction in tube potential from 140 to 80 kV leads to an estimated decrease of muscle area and SMI by approximately 5% and of steatotic muscle area of 13%. (Influence of tube potential on CT body composition analysis. Morsbach F, Zhang YH, Nowik P, Martin L, Lindqvist C, Svensson A, Brismar TB.Nutrition. 2018 Sep;53:9-13. doi:10.1016/j.nut.2017.12.016. Epub 2018 Feb 6.) Use of contrast media will also affect the measurements, overestimating average SMI of about 3% and underestimating steatotic muscle area of 13%. (Body composition evaluation with computed tomography: Contrast media and slice thickness cause methodological errors. Morsbach F, Zhang YH, Martin L, Lindqvist C, Brismar T.Nutrition. 2019 Mar;59:50-55. doi: 10.1016/j.nut.2018.08.001. Epub 2018 Aug 9). Also slice thickness will affect the measurements with 2% smaller SMI when obtained from 10 mm thick slices compared to that of 2 mm thick (Morsbac Nutrition 2019).
“However, it should be noted that radiological evaluation does not reflect the strength.” This should be referenced. (“evaluation” should probably be “assessment”).
“Evaluation of sarcopenia is crucial in patients with chronic liver disease…” references needed
Table 2: Improve the text in the column “adverse effects”
Table 2. Although Carias et al stated that there was a trend towards worse survival in patients with sarcopenic obesity it was by far not statistically significant. To be a trend it has to have a p-value <0.10 and the material has to be rather large (several hundreds, if not thousand). Carias’ material had neither.
Table 2 Fujiwara: “Sarcopenia (HR = 1.52, p = 0.001) add poor muscle quality (HR = 1.34, p = 0.020) were… “ typing error “add” should be “and”.
Reviewer 2 Report
The present review article contains radiological approaches of sarcopenia, which is characterized by the age-related decline of skeletal muscle plus low muscle strength and/or physical performance, especially in chronic liver disease. I think it was easy to understand, simple and well organized article. However, additional explanation is needed for some important points.
1. References and explanations for the accuracy and complementation methods for functional muscle strength and/or physical performance which is one of the crucial parts of sarcopenia definition in each radiological method should be added.
2. In each method, more references and explanations for the cut-offs performed in patients with chronic liver disease and the resulting clinical outcome should be added.
3. In particular, analysis and/or references according to HBV, HCV, NASH, etc., which are the main causes of chronic liver disease, and ethnic differences should be added.
4. It is necessary to consider whether each test method affects in accuracy depending on the status of chronic liver disease, liver function and/or liver cirrhosis.
5. In addition, it is likely that USG or DEXA will be used rather than CT or MRI in patients with chronic liver disease in real clinical situation, so it would be very helpful if you could provide a recommendations of those methods.
Round 2
Reviewer 1 Report
The authors have responded well to almost all my comments, but as I pointed out in my previous report the manuscript is full of subtle linguistic and grammatical errors. Those are almost impossible to correct unless being native speaking and understanding medical research. I will not point out all the mistakes but I give a few examples below showing how difficult it is to find those:
- Appendiceal means ”around appendix” - the authors mean appendicular.
- ”Primary sarcopenia is a naturally occurring phenomenon with aging; however, when its severity is beyond what can be justified by ageing in people with chronic illness, it is called secondary sarcopenia [2].”
In this sentence it does not make sense to say ”… what can be justified by ageing in people with chronic illness” (ie beyond the sum of ageing and the chronic disaese). The authors more probably mean ”…when its severity is due to chronic illness beyond what can be justified by ageing alone, it is called secondary sarcopenia”. These errors make the manuscript less enjoyable to read and it does not make the effort done justice. - ”implemented with” should probably be ”adjusted for”
I strongly recommend consulting a professional, native copy editor used to medical research. Consulting a colleague, or even Chinese born English/American citizen will not solve the problem as there are simply too many subtle errors requiring professional editing. Those error make the manuscript cumbersome to read. It is well worth the investment – your paper will be read and cited more if it is easy to read.
The authors have inserted the requested images - good! Regarding those:
Fig 3 State that M psoas has been coloured red in panel c and d (”c” has been horisontally reversed) and all muscle in panel e and f.
Fig 4 includes the wrong images. I guess the authors have done some extreme windowing to the small detailed inserted images. This makes the left kidney appear fatty – that cannot be correct. The spinal fluid is also white on the ”fat” image. The large image is probably a T2 Haste (which can only be used for drawing the contour, not for semiautomatic quantification) and I guess that the inserted ones simply are the T2 images after extreme windowing. Ask your radiologist for the in-phase and out of phase images or ”water only” and ”fat only” images from the Dixon imaging and ask the radiologist to apply proper windowing. Then make a figure with 4 panels instead of the detail insert. The pulse sequences used should also be stated.
Regarding the new text:
Line 119 I do not understand what is meant by "the umbilicus layer". The explanation can also be better.
Line 283: The correct paper has not been cited regarding kV. Morsbach et al showed the importance of kV in the paper Influence of tube potential on CT body composition analysis. Morsbach F, Zhang YH, Nowik P, Martin L, Lindqvist C, Svensson A, Brismar TB.Nutrition. 2018 Sep;53:9-13. doi:10.1016/j.nut.2017.12.016. Epub 2018 Feb 6.
In their 2019 paper they showed the effect of CM on SMI (as already correctly stated by the authors on line 284 and 285). Simply insert a ref to the 2018 paper on row 283.
